# Small All-Range Lidar for Asteroid and Comet Core Missions

**DOI:** 10.3390/s21093081

**Published:** 2021-04-28

**Authors:** Xiaoli Sun, Daniel R. Cremons, Erwan Mazarico, Guangning Yang, James B. Abshire, David E. Smith, Maria T. Zuber, Mark Storm, Nigel Martin, Jacob Hwang, Jeff D. Beck, Nathan R. Huntoon, Dick M. Rawlings

**Affiliations:** 1NASA Goddard Space Flight Center, Greenbelt, MD 20771, USA; daniel.cremons@nasa.gov (D.R.C.); erwan.m.mazarico@nasa.gov (E.M.); guangning.yang-1@nasa.gov (G.Y.); james.b.abshire@nasa.gov (J.B.A.); 2Department of Astronomy, University of Maryland, College Park, MD 20742, USA; 3Department of Earth, Atmospheric and Planetary Sciences, Massachusetts Institute of Technology, Cambridge, MA 02139, USA; smithde@mit.edu (D.E.S.); zuber@mit.edu (M.T.Z.); 4Fibertek Inc., Herndon, VA 20171, USA; mstorm@fibertek.com (M.S.); nmartin@fibertek.com (N.M.); jhwang@fibertek.com (J.H.); 5Leonardo DRS Electro-Optical Infrared Systems, Dallas, TX 75243, USA; jeff.beck@drs.com (J.D.B.); nathan.huntoon@drs.com (N.R.H.); dick.rawlings@drs.com (D.M.R.)

**Keywords:** lidar, remote sensing, pseudo-noise code

## Abstract

We report the development of a new type of space lidar specifically designed for missions to small planetary bodies for both topographic mapping and support of sample collection or landing. The instrument is designed to have a wide dynamic range with several operation modes for different mission phases. The laser transmitter consists of a fiber laser that is intensity modulated with a return-to-zero pseudo-noise (RZPN) code. The receiver detects the coded pulse-train by correlating the detected signal with the RZPN kernel. Unlike regular pseudo noise (PN) lidars, the RZPN kernel is set to zero outside laser firing windows, which removes most of the background noise over the receiver integration time. This technique enables the use of low peak-power but high pulse-rate lasers, such as fiber lasers, for long-distance ranging without aliasing. The laser power and the internal gain of the detector can both be adjusted to give a wide measurement dynamic range. The laser modulation code pattern can also be reconfigured in orbit to optimize measurements to different measurement environments. The receiver uses a multi-pixel linear mode photon-counting HgCdTe avalanche photodiode (APD) array with near quantum limited sensitivity at near to mid infrared wavelengths where many fiber lasers and diode lasers operate. The instrument is modular and versatile and can be built mostly with components developed by the optical communication industry.

## 1. Introduction

Lidar (light detection and ranging) instruments have been used to measure surface elevation, shape, tidal deformation, spin axis, and reflectance in planetary missions [1,2,3,4]. Lidars have become essential tools for studying small planetary bodies, such as asteroids and comet cores, for determining the shape, rotational and orbital dynamics, and surface structure and interior properties along with gravity measurements [5,6,7,8,9]. Asteroids and comet cores are generally much smaller than major planets but they tend to have irregular shapes. Combined with their low-gravity, the spacecraft orbit configurations for different science observations often result in a wide range of orbit altitudes above the surface. Therefore, a lidar for missions to small bodies needs to accommodate a much wider dynamic range of the target distances than previous planetary lidars that were used in near-circular mapping orbits. The lidar should also support other mission operations, including reconnaissance from a long distance and landing or sample collection at close distances. Conventional orbiting lidar, such as those described in [3], do not have the needed operational flexibility and dynamic range. Here we describe a Small All-range Lidar (SALi) designed specifically for missions to small planetary bodies. It uses a fiber laser with return-to-zero pseudo-noise (RZPN) code laser modulation, correlation detection, and a pixelated linear mode detector that is sensitive to single photons. The lasers can be built at lower cost by leveraging technologies from the optical fiber communication industry. The combination of these new technologies collectively represents a new class of planetary lidar well suited for missions to small planetary bodies.

Small planetary bodies are remnants of planetary formation. They preserve constituents that are indicative of conditions in the early Solar System in mostly pristine settings. Numerous astronomical surveys of asteroids [10,11,12] have provided information about their population and implications for the evolution of the solar system [13,14]. The near-Earth object (NEO) population also poses a threat of Earth impact. A better understanding of their internal structure can help develop mitigation strategies. Several space missions have flown to asteroids and comets for extensive surveys from orbit [15,16,17,18,19,20,21,22]. The lidars on Hayabusa2 and OSIRIS-REx were essential science instruments for measuring the shape, gravity, and morphology of the asteroids to help understand their internal structure and their formation. The shape data collected by the lidars were also essential for spacecraft guidance, navigation, and control (GN&C) operation during touch down and sample collection.

Lidars for missions to small planetary bodies have different requirements than those for major planets because of the lower orbital altitudes and slower ground track speeds. They do not need to have high-power lasers and large receiver aperture sizes. The major technical challenges in the lidar design are its flexibility and dynamic range because of the irregular shape of the body and the wide range of orbit altitudes. The lidar should provide a coarse measurement of the shape and rotation axis of the body from a high altitude during the reconnaissance phase of the mission prior to high precision mapping. It is also highly desirable that the same lidar can provide real time spacecraft altitude and velocity data to the spacecraft GN&C system during descent and touchdown. A multi-beam or scanning lidar is preferred to improve the spatial coverage and provide surface slope and roughness measurements.

There have been several lidars flown to asteroids over the past 25 years [6,7,8,9]. The designs of those lidars were similar to previous planetary lidars but with reduced laser pulse energy and receiver telescope size. They all used diode pumped Q-switched Nd:YAG lasers with fixed output pulse energy, which limited the dynamic range of the lidar measurements. They were mostly designed for mapping the objects from a given orbit altitude. Separate lidars are used for GN&C during descent and touchdown, such as the Doppler lidars (NDL) [23,24] and hazard detection lidars (HDL) [25]. The Chang’E 4 lunar lander used a single beam lidar for GN&C at high altitude and a flash lidar for hazard detection at 100 m above the surface [26]. A flash lidar illuminates an area of the target with a laser pulse and uses a pixelated detector to provide a 3-D image of the target from that single laser pulse. Flash lidars are well suited for hazard avoidance but cannot range over long distances because the laser pulse energy is spread out over the entire scene. The OSIRIS-REx mission carries both a science mapping lidar [9] and a flash HDL [27]. The mapping lidar provided a detailed 3-D model of the asteroid which was used in combination with the camera images for the precision touchdown on asteroid Bennu even without the HDL. A simple laser range finder is still desired in future to provide the spacecraft range and velocity in real time during descent and touchdown.

SALi is specially designed for missions to small planetary bodies. It features three new technologies. First, a RZPN code is used to modulate the laser transmitter. The receiver detects the laser pulse pattern instead of individual laser pulses by correlating the received signal with the RZPN kernel [28]. This technique enables the use of low peak-power lasers for long-distance ranging without aliasing. It uses a patented ternary (i.e., −1, 0, 1) RZPN kernel [28], which significantly reduces the effects of background noise and the detector dark noise compared to conventional pseudo noise (PN) lidar. Second, SALi uses optical fiber lasers developed by the telecommunications industry. The laser output power can be continuously adjusted over a wide range according to the target range. Third, it uses a new linear mode photon counting HgCdTe avalanche photodiode (APD) array with near quantum limited sensitivity from visible to mid-infrared wavelength [29], which maximizes the receiver sensitivity and gives long-distance ranging capability. The APD gain can be adjusted from unity to above 1000 with little excess noise and the APD array gives linear analog outputs over a wide dynamic range.

SALi is designed to operate in three modes: a coarse survey mode from a long distance, a precision mapping mode from a mid-altitude orbit, and a real-time GN&C sensor mode for touchdown and sample collection. The primary science measurements are conducted in the mapping mode. The multi-beam measurements from different orbits create large number of cross-over measurement points which can be used to refine the spacecraft orbit position and laser pointing angles [30] and construct a global shape model of the body with high geodetic accuracy. The repeated observations of the same locations at multiple times enables monitoring of the body orientation, spin state, spin variations due to the solar radiation effect, response to recent planetary encounters, and degassing-related torques for active comet cores.

This paper is organized as follows. Section 2 describes the PN and RZPN lidar technique and the particular RZPN code selected for SALi. Section 3 describes the instrument and major subsystem design, including the laser, the optics, the detector, digital signal processing electronics, and the expected lidar performance. Section 4 gives a short conclusion.

## 2. RZPN Lidar Technique

### 2.1. PN Code Lidars

PN codes have long been used for radio frequency (RF) ranging systems, such as in the Global Positioning System (GPS). PN code lidars have previously been used for atmospheric profiling and ranging measurements [31,32,33,34]. A PN code has the advantageous property that it has no apparent correlation with other binary sequences or itself unless it is perfectly aligned in time [35]. In a PN code lidar, the laser is modulated by a binary PN code sequence. The received signal is cross-correlated with a bi-polar form of the code, called the kernel, which is equal to +1 for binary code value “1” and −1 for binary code value “0”. The number of binary bits in a PN code sequences is given by $N_{b}=2^{m}-1$ with m an integer. There are $\left(N_{b}+1\right)/2$ ones and $\left(N_{b}-1\right)/2$ zeros in the sequence. The normalized cross correlation of the code and the kernel is equal to unity when the code and kernel line up bit by bit and equal to $2/\left(N_{b}+1\right)$ everywhere else. The ones and zeros occur at random statistically, thus the term pseudo-noise and the useful correlation properties. Unless the PN code matches the kernel in pattern and time, the kernel effectively randomizes the received signal and background noise by multiplying them by +1 or −1 at random and cancels them out upon correlation. The correlation function peaks when the input signal matches the kernel. The peak value of the correlation function is equal to the total number of received signal photons, which can be used to calculate the surface reflectance. The peak location in time gives the signal delay or the laser pulse time-of-flight from which the target range can be determined. The ranging precision can be improved by increasing the sampling rate of the received signal such that there are multiple samples per received laser pulse waveform [36,37]. The correlator output is periodic and hence there is range ambiguity which is determined by the code length.

Use of PN code modulation enables lidar to use lasers with lower peak powers for long distance ranging. The receiver sensitivity increases with the received integration time, which allows a tradeoff between the maximum range and measurement rate. A major shortcoming of a conventional PN code lidar is its susceptibility to background illumination and detector dark noise. The receiver for a conventional PN code lidar integrates the background and detector dark noise with the signal for the entire receiver integration time, which is often the dominant noise source.

### 2.2. RZPN Code Lidars

We have developed a variant of the PN code and kernel, which retains all the advantages of a regular PN lidar but is much less susceptible to background noise [28]. In this approach, the laser is only on at the beginning of the PN code bit interval for a short time and off, hence return-to-zero, for the rest of the bit period. The RZPN kernel takes the same values as the PN code kernel but returns to zero for the reminder of the bit time, as shown in Figure 1. The RZPN lidar technique has been successfully demonstrated in the lab and the field with a breadboard instrument [37,38].

There are three major advantages to the RZPN lidar technique. First, it nulls out all background noise in the return-to-zero time period when calculating the correlation function. Second, it uses shorter laser pulse width which helps to improve the ranging precision. Third, it provides space in the remaining bit intervals for additional measurement channels without direct interference as long as they use different RZPN codes. This laser modulation technique provides a new degree of freedom in PN code lidar design, namely, the laser pulse width. This approach is especially suited for fiber lasers where the pulse width and pulse rate can be varied over a wide range under the same average laser output power and the electrical power. The average output power can also be adjusted by changing the pump laser power. Fiber lasers have a much higher electrical-to-optical conversion efficiency than Nd:YAG lasers. They are small and modular, and widely used in the telecommunication and other industries. They have advanced rapidly in recent years.

### 2.3. RZPN Code Detection

Assuming a rectangular pulse shape, a transmitted laser signal in a RZPN code lidar over one code length period can be expressed as
(1)$$x\left(t\right)=E_{t}\overset{N_{b}-1}{\underset{i=0}{\sum }}a_{i}u\left(t-iT_{b}\right)$$
where $E_{t}$ is the laser pulse energy, $N_{b}$ is the number of bits in the underlining PN code, $a_{i}$ is the binary code sequence, equal to 1 or 0, $T_{b}$ is the bit interval, and $u\left(t\right)$, is the unit pulse function given by
(2)$$u\left(t\right)=\left\{\begin{array}{cc} \frac{1}{T_{p}}, & 0\leq t\leq T_{p}\\ 0, & else \end{array}\right.$$
with $T_{p}$ the laser pulse width. For RZPN code modulation, $T_{p}<T_{b}$. The RZPN code period is equal to the number of bits in the code times the bit interval. The laser pulse width, or pulse duty cycle, is a parameter to be optimized.

The received signal is the convolution of the transmitted signal and the target impulse response function, $h_{T}\left(t\right)$, which can be written as
(3)$$y\left(t\right)=\int_{0}^{\infty }h_{T}\left(t-\tau_{1}\right)x\left(\tau_{1}\right)d\tau_{1}.$$

The normalized correlation function of the received signal and the kernel is given by
(4)$$z\left(t\right)=\left(\frac{2}{N_{b}+1}\right)\times \left(\frac{T_{p}}{E_{t}}\right)\times \int_{0}^{T_{s}}y\left(\tau_{2}\right)k\left(\tau_{2}+t\right)d\tau_{2}$$
where $T_{s}=N_{b}T_{b}$ is the code length time period and $k\left(t\right)$ is the RZPN code kernel, which is a periodic function. Over one code length period the kernel can be written as
(5)$$k\left(t\right)=\overset{N_{b}-1}{\underset{i=0}{\sum }}{a^{\prime }}_{i}u\left(t-iT_{b}\right)$$
with
(6)$${a^{\prime }}_{i}=\left\{\begin{array}{cc} 1, & a_{i}=1\\ -1, & a_{i}=0 \end{array}\right..$$

Since the convolution and correlation are linear operations, their order can be exchanged and Equation (3) can be rewritten as
(7)$$z\left(t\right)=\int_{0}^{\infty }h_{T}\left(t-\tau \right)z_{0}\left(\tau \right)d\tau$$
with
(8)$$z_{0}\left(t\right)=\left(\frac{2}{N_{b}+1}\right)\times \left(\frac{T_{p}}{E_{t}}\right)\times \int_{0}^{T_{s}}x\left(\tau \right)k\left(\tau +t\right)d\tau .$$

Therefore, the cross-correlation function of a RZPN lidar is equal to the convolution of the target impulse response with the normalized cross-correlation function of the transmitted RZPN code with its kernel given in Equation (8). For a rectangular laser pulse shape, the peak of this cross-correlation function is a triangle with the base equal to two times the laser pulse width. It is a periodic function with the period equal to the PN code length. The impulse response of the target can be much wider than the laser pulses. It broadens the shape of the peak of the correlation function, but it does not cause aliasing.

In practice, the received signal is digitized and so is the kernel. Assuming the signal is digitized at interval $\Delta t_{s}$, the total number of data points within a PN code period becomes
(9)$$N_{s}=N_{b}\left(T_{b}/\Delta t_{s}\right).$$

As mentioned earlier, the signal is sampled at a fraction of the laser pulse width, i.e., $\Delta t_{s}\ll T_{b}$. As a result, the number of data points to process for a RZPN lidar is much larger than that of a conventional PN lidar.

For relatively weak signals, the signal or the correlation results need to be averaged to achieve a sufficient signal-to-noise ratio (SNR). The order of averaging and correlation can be exchanged without affecting the result since they both are linear operations. In practice the signal is usually averaged first, since signal averaging can be carried out relatively easily and it reduces the data volume to be carried forward.

Finally, a peak search is performed on the outcome of the cross-correlation function. The centroid of the peak gives the target distance, the width of the peak gives an indication of the laser pulse spreading by the target slope and roughness, and the integral of the peak gives the received laser energy, which can be used to infer the surface reflectance.

### 2.4. Comparison to Other Lidar Techniques and Constrainst in RZPN Lidar Applications

The RZPN lidar technique enables the use of low peak power lasers for long distance ranging. The efficiency of a PN or RZPN lidar in terms of the average transmitted laser power required to achieve a given receiver SNR is less than a single pulse detection lidar [39], but it enables the use of more efficient lasers, such as fiber lasers. Thus, the overall instrument efficiency can be significantly higher.

PN and RZPN lidars are generally far less susceptible to laser speckle noise than conventional lidars. Laser speckle can be a significant noise source for conventional lidar using small laser footprint sizes and low pulse rates [40]. Laser speckle noise still exists in RZPN lidars but it becomes negligible compared to other noise after averaging a large number of laser pulses during the signal processing.

RZPN code modulation has an additional benefit of allowing time division multiplexing of other measurement channels during the return-to-zero time periods. For example, one can insert another measurement channel at a different laser wavelength with a different RZPN code to measure the differential spectral absorption feature of the reflected laser light [38].

The major technical challenge of RZPN lidars is the more sophisticated signal processing at the receiver. The receiver has to multiply the received signal with the kernel term by term and sum all the terms to obtain a single value of the cross-correlation function. The computation has to be repeated for every time shift of the cross-correlation function within the code length. Because of the narrow laser pulse width and the high signal sampling rate, the total number of computation steps required for a RZPN lidar can be difficult to carry out in real time. However, these computations are all simple and highly repetitive. The computations can be carried in parallel in hardware with available field programmable gate arrays (FPGAs). The correlation function can also be calculated in the frequency domain with the use of Fast Fourier Transform (FFT) and inverse FFT, which reduces the number of computations by orders of magnitude.

Another limitation of RZPN lidars is the relatively long receiver integration time for a reliable measurement. During this time the target range needs to be constant, or the effect of the relative motion between the spacecraft and the target area needs to be properly compensated. This can be a challenge for measurements of major planets because of the relatively high orbit velocities of the spacecrafts. The Doppler shift due to the relative motion between the spacecraft and the target surface can be pre-compensated by adjusting the sampling clock frequency at the receiver based on the prediction. However, the small-scale surface topography cannot be predicted a priori, which can cause small time shifts in the received RZPN signal sequence and a reduction in the SNR. The residual changes in range rate in the orbit prediction can cause a time expansion or contraction in the received signal which causes a smear in the averaged signal and, in turn, the cross-correlation function. The smear can be reduced by shortening the signal averaging time, calculating the cross correlation at a higher rate, and averaging the correlation functions afterwards. However, this approach requires faster signal processing and more resources in the FPGA.

### 2.5. RZPN Lidar for Small Body Missions

RZPN lidar are best suited for missions to small planetary bodies, where the ground track speeds are relatively low and target range varies more slowly. Table 1 lists the rotation parameters and the sizes of the targets of most asteroid and comet missions to date. Based on flown or anticipated spacecraft trajectories in proximity of these bodies, it shows the residual range rate error in the orbit prediction for all small bodies is less than 10 m per second. When the receiver signal processing from each range measurement can be completed within 0.01 s, the total accumulated time error is 0.67 ns (0.1 m), which is well within the laser pulse width and should be acceptable. For large bodies like the dwarf planet Ceres, the signal processing would have to be completed much faster.

We have carried out a detailed computer simulation to study the effects of range rate uncertainty, local topography, orbital effects, and the receiver comparator circuit on the correlation results. In addition, we have designed and implemented a real-time, FFT-based correlation breadboard using a Xilinx FPGA and the SALi RZPN code. These results will be reported in a future publication.

## 3. SALi Instrument Design and Expected Performance

### 3.1. System Design

A prototype SALi instrument is currently being developed using the RZPN lidar technique. It combines the designs of an existing CubeSat laser communication system [48,49,50,51] and a CubeSat integrated detector cooler assembly (IDCA) [29,52], as depicted in in Figure 2. The instrument parameters are given in Table 2. The size and mass given in Figure 2 are preliminary estimates based on the two existing CubeSat subsystems with an aluminum structure and housing. Detailed instrument design is on-going and will be reported at a later time.

The SALi instrument uses a modular design. The primary focus of the current instrument development effort is the integration of the HgCdTe APD IDCA and real time RZPN signal processing with the FPGA. Other subsystem designs are relatively mature and they can be improved at a later time. For example, beryllium can be used instead of aluminum for the instrument chassis and support structure to improve the thermal conductance and reduce mass. A more efficient fiber laser assembly can be used to reduce the electrical power. A larger receiver telescope can also be used to extend the maximum measurement range.

There are three instrument operation modes: the long-distance survey mode, the mid-altitude mapping mode, and the descent and landing mode. Table 3 lists the instrument parameters for each mode. In the survey mode, all the 16-pixel detector outputs are summed together and the receiver integration time is set to 10 s (0.1 Hz) to achieve ranging at >500 km. In this mode, the lidar acts as a single beam lidar to provide a coarse measurement of the shape and rotation axis of the body. In the mapping mode, all 16 pixels operate independently and provide a 2 × 8 pixel swath along the ground track. In the landing mode, the instrument is used as a G&NC sensor. The major difference during landing mode operation is that the laser is modulated with periodic pulses (similar to a flash lidar) instead of using the RZPN code. During landing, the instrument provides a real-time multi-pixel target range, spacecraft velocity, and ground surface slope measurements during descent and touchdown.

The parameter values of the RZPN code sequence are given in Table 4. The code was chosen to optimize the lidar performance for the mid-altitude mapping phase of the asteroid investigation. The laser pulse width is 8 ns, which matches the detector impulse response. The signal is sampled at about 1 GHz. The code length is chosen to give a 9.75 km period (range ambiguity) of the correlation function, which coupled with the spacecraft orbit knowledge can be used to determine the absolute target range. The laser illuminates the entire scene subtended by the 2 × 8 pixel receiver FOV. The signal from each pixel is processed independently but with the same RZPN kernel. The same RZPN code and signal processing algorithm is used for the long-rang (Mode 1) operation but the cross-correlation functions from all 16 receiver channels are summed together, which is equivalent to summing the detected photons from all 16 pixels as one to turn the instrument into a single beam lidar. The results are further averaged over the receiver integration time to achieve the required SNR before the peak search. The cross-correlation function is still calculated every 100 ms so that the timing misalignment due to residual uncompensated Doppler shift does not increase due to the longer receiver integration time. If the timing misalignment is still too large, the laser pulse width and the PN code bit time can be lengthened to give a larger tolerance for time misalignment.

### 3.2. Estimated Instrument Performance

The most important aspects of SALi performance are the probability of target detection, the precision of ranging, and the precision of surface reflectance measurement. The ranging accuracy depends the frequency stability of the on-board clock and the requirement can be met with a proper choice of the clock oscillator and time keeping between the instrument and the spacecraft.

The target is detected from a peak search of the correlation function. Because they contain a large number of signal and background photon counts summed over a relatively long integration time the peak and baseline of the correlation function can be approximated as Gaussian random variables according to the Central Limit Theorem. A matched filter can be used prior to the peak search to improve the SNR. The probability of correct detection can be written as
(10)$$P_{d}={\left[\int_{-\infty }^{\infty }e^{\frac{-{\left(z_{p}-\mu_{p}\right)}^{2}}{2\sigma_{p}^{2}}}\;\int_{-\infty }^{z_{p}}e^{\frac{-{\left(z_{b}-\mu_{b}\right)}^{2}}{2\sigma_{b}^{2}}}dz_{b}\;dz_{p}\right]}^{N_{pk}-1}$$
where $\mu_{p}$ and $\sigma_{p}^{2}$ are the mean and variance of the peak of the correlation function, $\mu_{b}$ and $\sigma_{b}^{2}$ are the mean and variance of the baseline of the correlation function, and $N_{pk}$ is the number of local peaks to be compared to find the highest one. The local peaks can be from the target return or noise, but the width of the local peaks is limited by the system electrical bandwidth [53], which in this case is limited by the laser pulse width according to Equation (8). Therefore, we can use $N_{pk}\approx N_{b}\left(T_{b}/T_{p}\right)$ when evaluating the probability of target detection.

The mean and variance of the peak of the correlation function with an ideal photon counting detector can be approximated as
(11)$$\mu_{p}=\langle {\dot{n}}_{s}\rangle \;T_{I}$$
and
(12)$$\sigma_{p}^{2}=\langle {\dot{n}}_{s}\rangle \;T_{I}+\left(\langle {\dot{n}}_{b}\rangle +\langle {\dot{n}}_{d}\rangle \right)\;\frac{T_{p}}{T_{b}}\;T_{I}$$
where $\langle {\dot{n}}_{s}\rangle$, $\langle {\dot{n}}_{b}\rangle$, and $\langle {\dot{n}}_{d}\rangle$ are the average rate of the signal photons, the background photons, and the detector dark noise count, respectively, and $T_{I}$ is the receiver integration time. It is shown in Equation (12) that a RZPN lidar reduces the effect of the background photons and the detector dark counts by the ratio of the laser pulse width to the PN bit interval. Thus, there is a clear advantage to minimize the width of the laser pulse while maintaining the same average signal photon rate.

The mean value of the baseline of the correlation function can be assumed to be zero for relatively long PN code. The variance of the baseline when the received laser pulses line up with the kernel in pulse intervals but not the bit pattern is given by
(13)$$\sigma_{b}^{2}=\langle {\dot{n}}_{s}\rangle \;T_{I}+\left(\langle {\dot{n}}_{b}\rangle +\langle {\dot{n}}_{d}\rangle \right)\frac{T_{p}}{T_{b}}\;T_{I}.$$

The variance when the received laser pulses fall within the return-to-zero portion of the kernel becomes
(14)$$\sigma_{b0}^{2}=\left(\langle {\dot{n}}_{b}\rangle +\langle {\dot{n}}_{d}\rangle \right)\frac{T_{p}}{T_{b}}\;T_{I}.$$

To simplify the calculation, we can use the worse-case variance given in Equation (12) for the entire baseline.

The ranging precision can be estimated as in conventional lidar [54], as
(15)$$\sigma_{R}\approx \frac{\sigma_{w}}{SNR_{p}}$$
with $\sigma_{w}$ the root-mean-square (rms) width of the laser pulse width and $SNR_{p}$ is the SNR of the peak of the correlation function given by
(16)$$SNR_{p}=\frac{\mu_{p}}{\sigma_{p}}=\frac{\langle {\dot{n}}_{s}\rangle \;T_{I}}{\sqrt{\langle {\dot{n}}_{s}\rangle \;T_{I}+\left(\langle {\dot{n}}_{b}\rangle +\langle {\dot{n}}_{d}\rangle \right)\;\frac{T_{p}}{T_{b}}\;T_{I}}}.$$

Figure 3 shows the calculated SALi performance for the nominal mapping operation at mid altitude and long-distance measurements. The calculations are based on the above model with the instrument parameters summarized in Table 2. For the mapping mode calculation, we assumed all 16 individual pixels operating independently at 0.1 s integration time. For the long-distance reconnaissance mode, we assumed that the signals from all the 16 receiver channels were summed together and the integration time was increased to 10 s. The surface reflectance (albedo) was assumed to be 10% at 1.55 μm wavelength, including the opposition effect [55]. The solar distance was taken as 1 AU for a hypothetical mission to a NEO. Pulse broadening due to surface slope is negligible due to the small instantaneous field of view (IFOV) and relatively wide laser pulse. We also assumed that the Doppler shift in the received signal has been properly compensated and its effects can be neglected.

### 3.3. Laser Transmitter and Optics

The design of the laser transmitter and the optics are nearly identical to those used in the Compact Laser Communication Terminal (CLCT) for free-space laser communication demonstrations [49,51]. The major differences are: (a) SALi uses a separate laser beam collimator (expander) instead of sharing the same telescope with the receiver telescope; (b) SALi does not use the laser pointing mechanism (fast steering mirror) used in CLCT; and (c) SALi uses a HgCdTe APD array detector with a cryo-cooler instead of the InGaAs photodiode in CLCT.

The laser transmitter consists of a seed laser and an erbium dopped fiber amplifier (EDFA) at 1.55 μm wavelength. It was originally designed for transmitting a 64-ary pulse position modulation (PPM) signal at up to 6-W average output power [56]. The laser pulse width, pulse rate, and output power are comparable to those required by SALi. For a small asteroid mission, 2-W average output power is sufficient. As the target distance gets closer, the laser output power can be reduced by at least a factor of 10 by adjusting the pump power of the fiber amplifier. The beam from the fiber laser amplifier is collimated and shaped into a rectangular pattern using a diffractive optical element (DOE) [57] to illuminate the entire scene subtended by the 2 × 8 pixel HgCdTe APD array. Each pixel has an IFOV of 60 × 60 μrad. Therefore, the laser beam has a divergence angle of 120 × 480 μrad. The spatial resolution is the product of the IFOV and spacecraft altitude. A small light deflector is built in at the edge of the laser collimator to scatter a small amount (<1%) laser energy off-axis to allow target range measurement down to <2 m when the transmitted laser beam and the receiver FOV no longer completely overlap due to the bistatic design.

The receiver telescope, shown in Figure 4, is designed to fit in a 1-U CubeSat volume and features an afocal off-axis Cassegrain reflector with an 8× magnification. To provide the largest possible aperture in the available volume the primary mirror has an effective numerical aperture of 1.3. The protected-gold-coated diamond-turned aluminum mirrors are integral to the all-aluminum structure, so no thermal metering is required. The aft optics consists of an optical bandpass filter and a set of focusing lenses to set the 60 × 60 μrad IFOV. The entire telescope structure is thermally stable and has already been space-qualified [51].

In the current SALi design, the focus of the telescope and aft optics is fixed to an intermediate distance such that it can give a near optimal performance when mapping of the asteroid or the comet core but still acceptable performance when becoming out of focus at longer target range. During the spacecraft touch-and-go (TAG) maneuver at close up distance, the out of focus can become so severe that the IFOVs subtended by all pixels overlap and SALi becomes a single beam laser range finder. A focusing mechanism may be added in the aft optics assembly if a multipixel measurement is needed.

### 3.4. Detector Cooler Assembly 

The SALi detector assembly consists of a 2 × 8 pixel HgCdTe APD array [58,59] in a Dewar cooled by a small Stirling cooler [60]. The HgCdTe APD has a near 90% quantum efficiency over the 0.9 to 4.3 μm wavelength range. The APD gain is nearly deterministic gain and there is little excess noise added when the photoelectrons are multiplied. The APD gain can be adjusted from 1 to >1000. At the maximum gain the photocurrent can completely override the preamplifier electronics noise and single photon event can be detected.

The voltage outputs of the HgCdTe APD array are linear with the incident optical signal power. When there are multiple photons in a received laser pulse, the output becomes the linear sum of pulse waveforms from individual detected photons without any dead-time or other nonlinear effects. The APD has a factor of 100 linear dynamic range at a fixed APD gain. The APD gain can be adjusted by a factor of 1000. Together with the laser power adjustment, SALi can have at least six orders of magnitude dynamic range.

In the present detectors the pixels are 20 μm in diameter and have a 64 μm pitch in a 2 × 8 format. Larger pixel arrays with more pixels are possible in the future. The HgCdTe APD arrays have been tested for radiation damage and shown to be suitable for use in a typical planetary mission [61]. The use of HgCdTe APDs also enables the use of other type of laser at wavelengths from 0.9 to 4.3 μm.

The HgCdTe APD array operates at 110 K and a cryo-cooler is required. The cooler used in the current IDCA is a military off-the-shelf miniature Stirling cooler [60]. It is designed for operation in sub-orbital and harsh operation environments and has demonstrated a multi-year operation lifetime.

An IDCA with the same HgCdTe APD array and the cooler has already been developed for a demonstration on a CubeSat [52]. A microlens array is used on top of the APD array which effectively brings the detector fill factor to near 100%. The IDCA with the microlens array has been laboratory tested per NASA General Environment Verification Standard (GEVS). A more detailed description of the test results of the HgCdTe APD and the IDCA can be found in [29]. The IDCA for SALi is the same as the previous ones but with the compressor and the cooler rearranged (Figure 5). There is also a major improvement in the cold shield and cold filter design to reduce the stray thermal photons. The cold shield aperture is chosen to give a numerical aperture of 0.14 (f/7). The cold filter has two pass bands, one from 1.00 to 1.13 μm and the other from 1.40 to 1.70 μm, for use with 1.03 μm or 1.55 μm fiber lasers, which are the most common types of fiber lasers from the industry at present. The thermal background photons from the f/7 cold aperture over these two passbands are negligible compared to the detector dark noise. The cold filter passband can be fixed after the exact laser wavelength is known. The out-of-band rejection is >6 optical density (OD). The combination of the cold shield and the cold filters ensures that thermal photons through the cold aperture and stray thermal photons around the cold shield are much lower than the intrinsic dark counts of the detector. The net weight of the IDCA is about 0.80 kg. The total electrical power consumption is about 7 W with the detector chip at 110 K and the IDCA housing at room temperature.

### 3.5. Electronics

A major technical challenge for SALi is the receiver electronics that can process multi-channel RZPN signals in real time. Our previous RZPN lidar demonstrations post-processed the signal and kernel in software [37,38], which took minutes to process one measurement for a similar RZPN code and signal sampling rate. The SALi receiver needs to process 16 channel signals in real time with digital signal processing on a FPGA.

Figure 6 shows a block diagram of SALi. The laser and the associated electronic circuits for the laser transmitter are identical to those used in the laser communication demonstration [48,49,50,51]. The signal processing electronics is also the same but with the FPGA reprogrammed for the RZPN lidar signal processing. The system controller for command and data handling (C&DH) and the DC power converters shown in Figure 6 are also the same as those used in the CubeSat laser communication demonstration.

A clock generation circuit is added that generates the clock for the transmitted RZPN code and the receiver signal sampling clock. The clock for the received signal is locked to the transmitter clock but with an offset to compensate for the Doppler shift due to the relative motion between the spacecraft and the target body. The Doppler shift is estimated based on the predicted orbit dynamics. Both clocks are locked to an external reference clock, which is currently a temperature compensated crystal oscillator (TCXO) but can be replaced with a chip scale atomic clock (CSAC) [62], if needed.

The pulse waveforms from each of the 2 × 8 pixels are digitized before being processed by the FPGA. To reduce the electrical power consumption, a bank of comparators is used as 1-bit digitizers instead of regular 8-bit analog-to-digital converters (ADCs). The threshold levels for the comparators are dynamically adjusted according to the signal level. A single separate ADC is used to digitize the sum of all detector outputs to monitor the received signal level and control the comparator threshold, the laser power, and the detector gain. For weak signals, the detector gain is set to the highest and the thresholds are lowered to just above the noise floor such that the detector is in single photon counting mode. For strong signals, the detector gain is reduced and the thresholds are raised such that only multi-photon signal pulses are likely to be detected and background and detector dark noise are suppressed. The use of comparators causes quantization errors for strong signals, but the losses in SNR are more than compensated by the stronger signal and lower noise count rate.

### 3.6. FPGA Design

The FPGA is used to generate the RZPN code to modulate the laser transmitter and process the received signal to locate the target and calculate the laser pulse time of flight and average received laser power.

The PN code is generated using the standard technique with a series of shift registers and logic feedbacks [35]. The pulse width of the PN code signal is reduced to give the specified RZPN pattern. The signal is then amplified to drive the laser.

The received signal from each pixel is serially fed into a shift register bank which periodically shifts out the content to an accumulator (histogrammer), which accumulates the signal over the RZPN code period. The range bin size of the histogrammer is the same as the signal sampling interval and the integration time of the histogrammer is set to 0.1 s, or 1538 codes. The cross correlation of the accumulated signal with the kernel is carried out in the frequency domain using FFTs, complex multiplication, and IFFTs. A peak search is conducted to locate the target return and determine the ambiguous range, return pulse width and energy. There are 16 sets of signal accumulation circuits, one for each pixel. The cross-correlation is carried out in series using one set of FFT/IFFT circuits. For the long-distance survey operation, the cross-correlation functions from all 16 pixels are summed together and then averaged up to 10 s integration time before the peak search. For short distance measurement during descent and landing, the laser is modulated with periodic pulses at about 65 kHz instead of a RZPN code. The received signals are sampled and histogrammed the same way but the FFT/IFFT is skipped and the signal goes directly to the peak search.

The FPGA also processes the ADC signal and estimates the received signal and noise power and dynamically control the comparator threshold with the use of a standard control loop. The system controller dynamically adjusts the laser power and the APD gain via a software control loop based on the target return signal power and the background power.

We have implemented the receiver processing algorithm in a breadboard test system using National Instruments (NI) FPGA hardware and LabVIEW software. The RZPN code and kernel were generated in the FPGA and the signals were transmitted and received in a loopback fashion. The received signal was histogrammed and corelated with the kernel in frequency space using the FFT method described above. A detailed description of that work will be described in a separate publication. We have verified that the correlation operation for each pixel can be carried out in <1 ms for the RZPN code specified in Table 4 using a Xilinx KU060 FPGA. This allows the system to process the ranges for all 16 pixels in less than 10 ms, which meets the design criterion of a 100 Hz measurement rate.

## 4. Conclusions

A small all-range lidar (SALi) is described, which is specially designed for space missions to small planetary bodies. The lidar uses return-to-zero pseudo-noise (RZPN) code laser modulation and correlation detection, which enables the use of low peak-power fiber lasers for long-distance ranging without aliasing. A newly developed, 2 × 8 pixel linear mode photon-counting HgCdTe avalanche photodiode (APD) array is used as the detector which gives a near quantum limited receiver sensitivity from 0.9 to 4.3 μm wavelengths. The laser power and the APD gain can both be adjusted to give six orders of magnitude dynamic range, enabling measurements of range and surface reflectance from hundreds of kilometers to near the target surface. The lidar is designed to operate in three modes, including reconnaissance at a long distance, mapping from orbit, and real-time multi-spot range and velocity measurements during descent and touchdown. A prototype instrument is currently being built by leveraging designs of the laser, optics, and electronics originally developed for a free-space laser communication demonstration. The combination of the fiber lasers, RZPN code modulation, and a multi-pixel single photon sensitive infrared detector enables a new class of small planetary lidars that are well suited for space missions to asteroids and comets.

## 5. Patents

The development of the RZPN lidar technique resulted in a US Patent, 7,982,861 B2, 2011 [29], by Abshire and Sun, who are also authors of this paper.

## Figures and Tables

**Figure 1 sensors-21-03081-f001:**
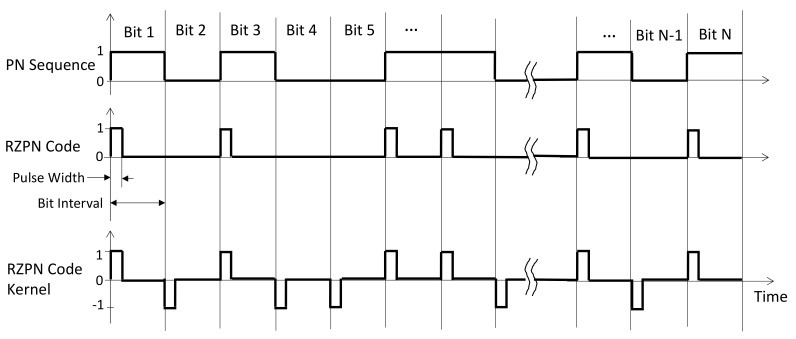
Schematic of a binary PN sequence (**top**), the corresponding RZPN code for the laser modulation (**middle**), and the kernel to be used for the correlation detection (**bottom**).

**Figure 2 sensors-21-03081-f002:**
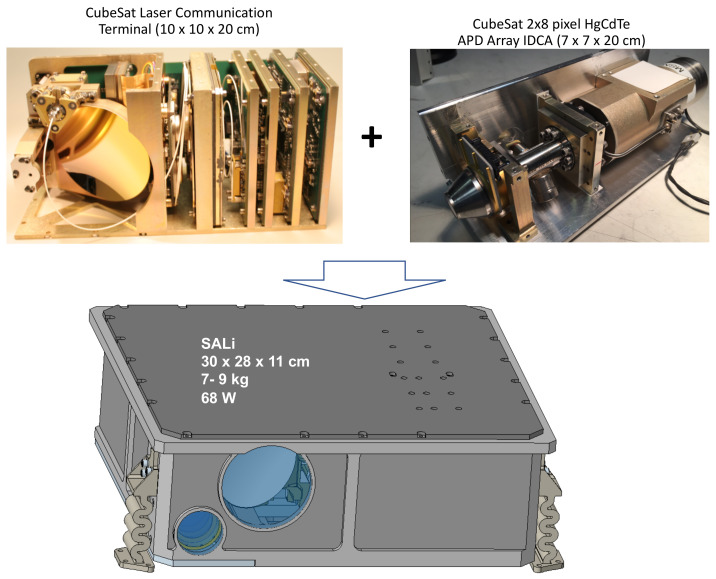
Conceptual design of the SALi instrument which combines a CubeSat laser communication terminal [50] with a HgCdTe APD detector-cooler assembly [29,52].

**Figure 3 sensors-21-03081-f003:**
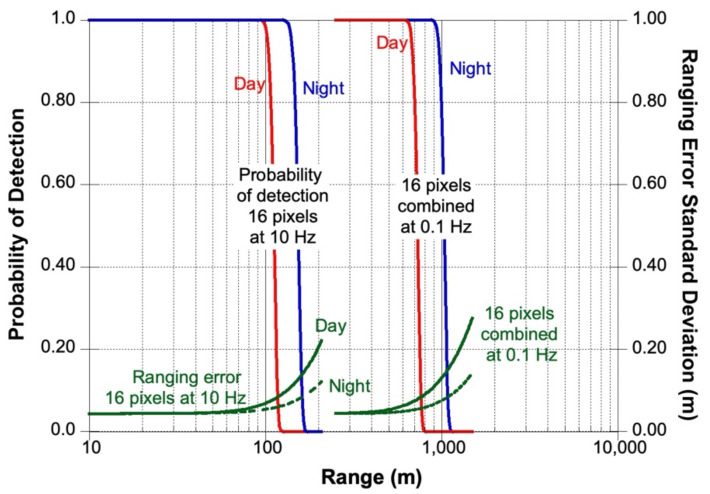
Calculated SALi ranging performance based on the measurement model above and instrument parameters given in Table 2.

**Figure 4 sensors-21-03081-f004:**
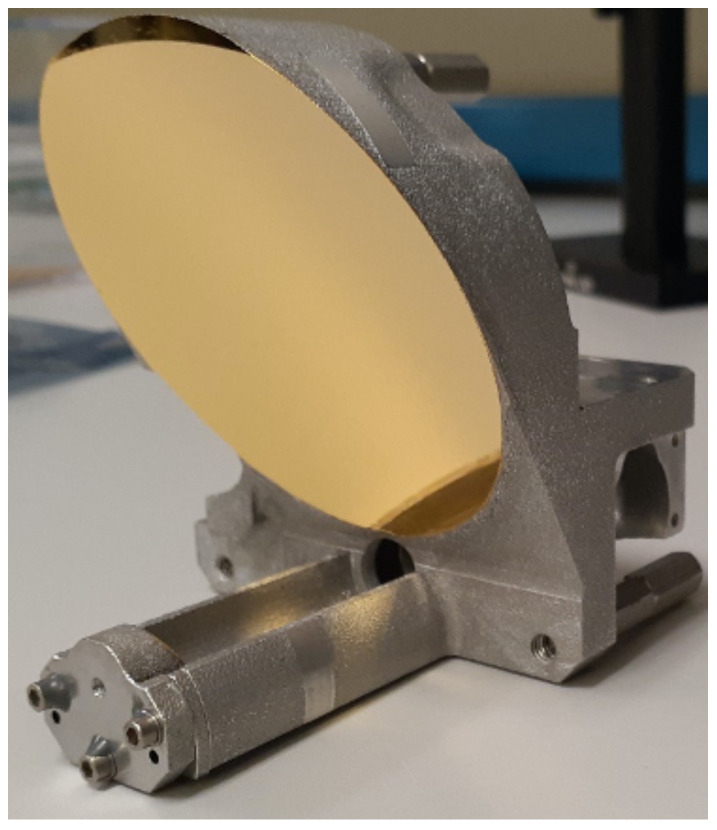
Photograph of the SALi receiver telescope. It features an afocal off-axis Cassegrain reflector with a clear aperture of 6.4 cm diameter and 8 times magnification.

**Figure 5 sensors-21-03081-f005:**
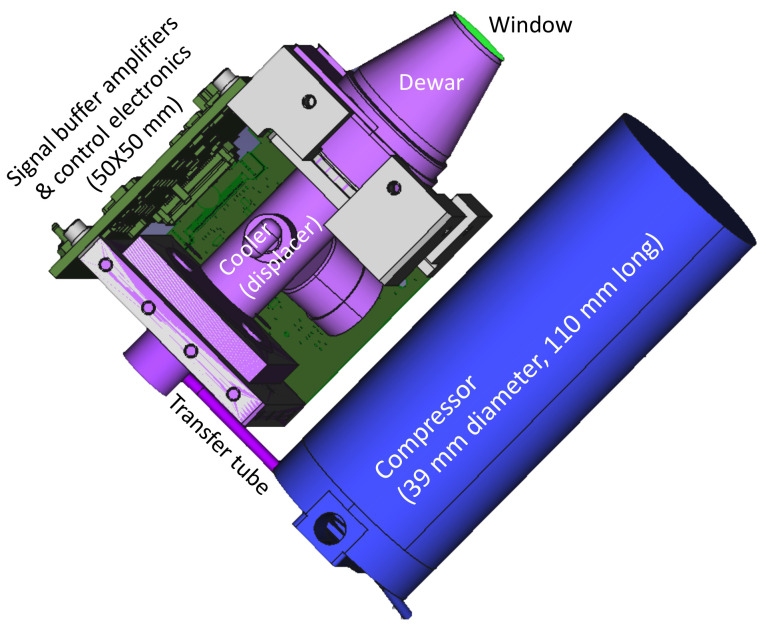
A computer-assisted design (CAD) drawing of the integrated detector cooler assembly (IDCA) for SALi.

**Figure 6 sensors-21-03081-f006:**
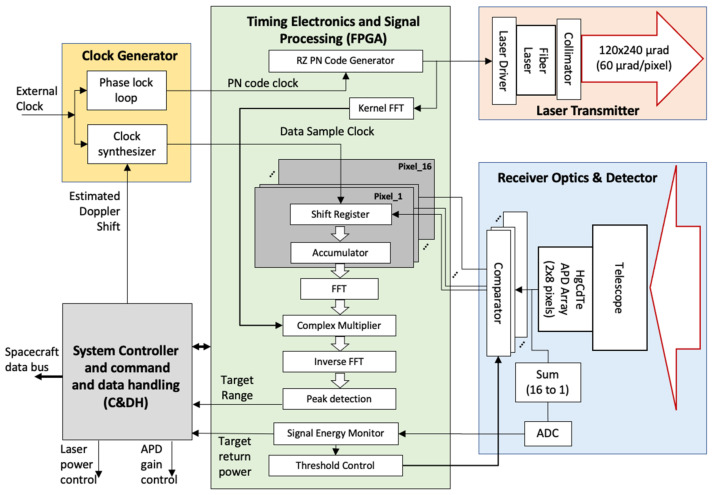
Block diagram of SALi instrument and electronics.

**Table 1 sensors-21-03081-t001:** List of orbit dynamics of past asteroids and comet missions. References are indicated for measured values. The other values are derived from shape and trajectory data.

Parameter	101955 Bennu	25143 Itokawa	162173 Ryugu	67P/Churyumov-Gerasimenko	433 Eros	Ceres
Mean Diameter	492 m ^1^	330 m ^1^	896 m ^3^	3.3 km ^5^	16.8 km ^1^	939.4 km ^1^
Rotation Period	4.3 h ^1^	12.1 h ^1^	7.6 h ^1^	12.8 h ^1^	5.3 h ^1^	9.1 h ^1^
Bond albedo	4.6% ^1^	27% ^2^	4.0% ^4^	6.2% ^6^	25% ^1^	9.0% ^1^
Topography	±35 m	±200 m	±50 m	±2 km	±10 km	±10 km
Reference Mission	OSIRIS-REx	Hayabusa	Hayabusa2	*CAESAR* ^7,8^	NEAR	*TBD* ^8^
Orbit Altitude	<1 to 7 km	7 to 20 km	0.1 to 20 km	2.5 to 10 km ^8^	20 to 200 km	30–70 km ^8^
Speed of ground track	2.5 to7 cm/s	<2 cm/s	<2 cm/s	10 to 50 cm/s ^8^	1.5 to 5 m/s	350 m/s ^8^
Orbit Prediction						
Residual Range Uncertainty	30 m	75 m	75 m	500 m ^9^	800 m ^9^	2 km ^9^
Residual Range Rate Uncertainty	10 cm/s	5 cm/s	10 cm/s	5 m/s ^9^	7.5 m/s ^9^	200 m/s ^9^

^1^ [41]; ^2^ [42]; ^3^ [43]; ^4^ [44]; ^5^ [45]; ^6^ [46]; ^7^ [47]; ^8^ notional (future mission); ^9^ After gross shape obtained.

**Table 2 sensors-21-03081-t002:** SALi instrument parameter values.

Instrument Parameters	Values
Average laser power	2.0 W
Laser wavelength	1.55 μm
Beam divergence	120 × 480 μrad
Transmit efficiency	90%
Receiver telescope	6.4 cm diameter
Receiver instantaneous field of view (IFOV)	60 × 60 μrad
Number of pixels	2 × 8
Receiver bandwidth	1.8 nm
Optical transmission	60%
Detector quantum efficiency	50%
Detector dark count	250,000/s
Estimated instrument size	29 × 28 × 11 cm
Estimated instrument mass Estimated power	9.1 kg (aluminum) 68 W

**Table 3 sensors-21-03081-t003:** SALi operation modes.

Lidar Parameter	Initial Survey	Mid-Altitude Mapping	Descent and Landing
Mode of operation	Single pixel coarse ranging	8 to 16 pixels precision ranging	16-pixel flash lidar
Range	500 to 50 km	50 to 10 km	10 km to 1 m
A priori uncertainty	±20 km	±200 m	±20 m
Maximum range rate uncertainty from orbit prediction	<0.25 m/s	<0.1 m/s	<0.5 m/s
Range rate due to topography	1 m/s	1 m/s	-
Measurement rate	0.1 to 1 Hz	1 to 100 Hz	10 to 100 Hz
Ranging precision and accuracy	10 m	0.2 m	0.05 m
Laser footprint diameter	<100 m from 500 km	6 m at 50 km	N/A
Surface reflectance accuracy	±20%	±5%	N/A

**Table 4 sensors-21-03081-t004:** RZPN code used in SALi.

Instrument Parameter	Value
PN code length, $N_{b}$	127 bits
Bit interval, $T_{b}$	512 ns
PN code period, $T_{s}=T_{b}N_{b}$	65,024 ns
Laser pulse width, $T_{p}$	8 ns
Pulse duty cycle, $T_{p}/\left(2T_{b}\right)$	0.78125%
Signal sampling interval, $\Delta t_{s}$	1 ns
Integration time, $T_{I}$	0.1 to 10 s

## Data Availability

Not applicable.

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
