# Peer review of "Small All-Range Lidar for Asteroid and Comet Core Missions"

_sensors, 2021, doi:10.3390/s21093081_

Round 1
Reviewer 1 Report
The presented paper covers a very interesting topic within the field of spaceborne lidar instrumentation. A new class of space lidar ist presented, its data are suited both for topographic mapping and as a base for sample collection and/or landing.
Irregular shape and low-gravity envorinment of small planetary bodies demand for high dynamic range of the lidar instrument. This is granted by using intensity-modulated fiber laser applying a return-to-zero pseudo-noise (RZPN) code.
The paper is well organized and written in a concise manner. However, it reads to some extent more like an extended Technical Report than a journal paper. The scientific contribution of this paper should to be highlighted a bit more in section 1. Moreover, could you please add the structure of the remainder of the article at the end of section 1?
Author Response
We appreciate your general comments about the paper and hope this revised version can be accepted for publication. We added a short segment in Lines 108-115 to further highlight the science contribution by this new instrument. We also added a few sentences at the end of the Introduction to give the structure of the reminder of the article.
Reviewer 2 Report
From my personal point of view, the work presented in this paper manuscript is interesting and has also well presented.
Author Response
Thank you! We made some minor editorial changes to the manuscript and hopefully further improved its clarity.
Reviewer 3 Report
In this paper , a new Small All-range Lidar (SALi ) in NASA was introduced. SALi instrument design is modular with a return-to-zero pseudo-noise (RZPN) code. And it integrated the HgCdTe APD IDCA and real time RZPN signal processing with the FPGA.
This is a new sensor and I wish it will be onboard for Asteroid and Comet Core Missions in the near future. I think the paper is deserved to be published.
In the paper ,the fiber laser were mentioned but there is little introduction. The authors should supplement the introduction about the laser itself.
Author Response
We added a few sentences in Lines 175 to 177 to highlight the advantage of the use of fiber lasers. Fiber lasers at 1550 nm wavelength are commercially available and have flown in space in recent years. A more detailed description of the particular fiber laser we used in this instrument is given in Section 3.3 and Reference [49] and [51]